# Low-Dose SARS-CoV-2 S-Trimer with an Emulsion Adjuvant Induced Th1-Biased Protective Immunity

**DOI:** 10.3390/ijms23094902

**Published:** 2022-04-28

**Authors:** Hung-Chun Liao, Wan-Ling Wu, Chen-Yi Chiang, Min-Syuan Huang, Kuan-Yin Shen, Yu-Ling Huang, Suh-Chin Wu, Ching-Len Liao, Hsin-Wei Chen, Shih-Jen Liu

**Affiliations:** 1National Institute of Infectious Diseases and Vaccinology, National Health Research Institutes, Miaoli 35053, Taiwan; liaohc@nhri.edu.tw (H.-C.L.); wuawan0522@gmail.com (W.-L.W.); cycjen@nhri.edu.tw (C.-Y.C.); madge.h@nhri.edu.tw (M.-S.H.); shenky057@nhri.edu.tw (K.-Y.S.); huangulin@nhri.edu.tw (Y.-L.H.); chinglen@gmail.com (C.-L.L.); chenhw@nhri.edu.tw (H.-W.C.); 2Institute of Biotechnology, National Tsing Hua University, Hsinchu 300044, Taiwan; scwu@life.nthu.edu.tw; 3School of Dentistry, Tri-Service General Hospital and National Defense Medical Center, Taipei 11490, Taiwan; 4Graduate Institute of Biomedical Sciences, China Medical University, Taichung 404333, Taiwan; 5Graduate Institute of Medicine, College of Medicine, Kaohsiung Medical University, Kaohsiung 80708, Taiwan

**Keywords:** SARS-CoV-2, vaccine, adjuvant, SWE

## Abstract

During the sustained COVID-19 pandemic, global mass vaccination to achieve herd immunity can prevent further viral spread and mutation. A protein subunit vaccine that is safe, effective, stable, has few storage restrictions, and involves a liable manufacturing process would be advantageous to distribute around the world. Here, we designed and produced a recombinant spike (S)-Trimer that is maintained in a prefusion state and exhibits a high ACE2 binding affinity. Rodents received different doses of S-Trimer (0.5, 5, or 20 μg) antigen formulated with aluminum hydroxide (Alum) or an emulsion-type adjuvant (SWE), or no adjuvant. After two vaccinations, the antibody response, T-cell responses, and number of follicular helper T-cells (Tfh) or germinal center (GC) B cells were assessed in mice; the protective efficacy was evaluated on a Syrian hamster infection model. The mouse studies demonstrated that adjuvating the S-Trimer with SWE induced a potent humoral immune response and Th1-biased cellular immune responses (in low dose) that were superior to those induced by Alum. In the Syrian hamster studies, when S-Trimer was adjuvanted with SWE, higher levels of neutralizing antibodies were induced against live SARS-CoV-2 from the original lineage and against the emergence of variants (Beta or Delta) with a slightly decreased potency. In addition, the SWE adjuvant demonstrated a dose-sparing effect; thus, a lower dose of S-Trimer as an antigen (0.5 μg) can induce comparable antisera and provide complete protection from viral infection. These data support the utility of SWE as an adjuvant to enhance the immunogenicity of the S-Trimer vaccine, which is feasible for further clinical testing.

## 1. Introduction

The recent vaccines have contributed to controlling the spread of COVID-19, especially in lowering the incidence of severe COVID-19 cases or mortality [1,2,3]. However, these vaccines, especially mRNA, are unstable at room temperature and sensitive to sunlight. The requirement of maintaining ultralow freezing temperatures during storage and transportation poses a huge limitation in less-developed countries. According to the simultaneous statistics from Our World in Data, only 10.6% of people have received at least one dose in low-income countries [4]. The development of a SARS-CoV-2 subunit vaccine has indispensable advantages for improving vaccination coverage to reach the herd immunity threshold; for example, it is safe, widely used for many viral diseases, a well-established manufacturing platform, and relatively stable for distribution to low- and middle-income nations.

Based on previous studies of SARS-CoV-1 and MERS-CoV, in the development of a COVID-19 vaccine, the spike protein was utilized as a prime antigen to generate protective immune responses. The spike is the major glycoprotein on the surface of coronaviruses and mediates viral attachment, fusion, and entry into host cells. Obviously, the spike is a crucial target antigen for rational vaccine design to induce neutralizing antibodies. In addition, the SARS-CoV-2 spike is a dynamic protein that contains two functional subunits. The S1 subunit harbors a receptor-binding domain (RBD) that recognizes the human angiotensin-converting enzyme 2 (hACE2) receptor, and the S2 subunit is responsible for the fusion of the viral and cellular membranes. While S binds to the receptor, host protease processing at the boundary between the S1/S2 subunit (furin cleavage site) and the S2′ site located upstream of the fusion peptide triggers an irreversible conformational change of the spike from a transient prefusion form to a highly stable postfusion state, facilitating membrane fusion [5]. To achieve the purpose of eliciting neutralizing antibodies, some approaches have been developed to stabilize the S prefusion form to preserve neutralizing epitopes, and these approaches include (1) abrogation of the furin cleavage site (residues 682–685) for protease resistance [6,7], (2) replacement of two prolines at positions 986 and 987 to help the epitope remain locked into the prefused conformation [8,9], and (3) C-terminal fusion with a trimerization motif [10,11]. These modifications have been applied to the currently available vaccines, such as BNT162b2 (Pfizer/BioNTech) [12], mRNA-1273 (Moderna) [13], NVXCoV2373 (Novavax) [14], and Ad26.COV2. S (Jannsen) [15].

Immunol stimulators are important for subunit vaccines, which lack pathogen-associated molecular patterns (PAMPs) to simulate innate immunity via pattern recognition receptors (PRRs); however, to overcome the limitations of subunit vaccines, adjuvants such as aluminum hydroxide, CpG-1018 [16,17], AS03 [11,18], and MF59 [10,19] have been employed to improve the immunogenicity of the SARS-CoV-2 spike in animal studies and clinical trials. Although aluminum hydroxide is a commercially available adjuvant, it only induces a modest immunity to the spike and shows less antigen-sparing capacity compared with the proprietary adjuvants CpG-1018 (Dynavax), AS03 (GlaxoSmithKline), and MF59 (Novartis). However, access to intellectual property rights will increase vaccine production costs, which may be a major barrier to mass vaccination in less-developed countries. Therefore, we evaluate a formulation of a recombinant S-Trimer with an MF59-like adjuvant, Sepivac SWETM (manufactured by SEPPIC), which was developed by the Vaccine Formulation Laboratory for technology transfer in open access [20,21]. SWE is an oil-in water emulsion with a similar composition to MF59, which consisted of squalene and two surfactants, Tween 80 and Span 85. SWE has been successfully paired with various vaccine candidates against Poliovirus [22], PRRSV [23], and H7N9 [24] in preclinical studies.

In order to generate cost-effective, long-lasting, and broad immune responses against evolving SARS-CoV-2 variants, this study is to assess the protective effects of the antigen doses and adjuvants formulated with S-Trimer vaccine. The cross-neutralizing antibodies against the SARS-CoV-2 Beta and Delta strains from different lineages were analyzed. As COVID-19 gradually goes from being an acute to endemic disease, such as seasonal influenza, the high safety subunit vaccine has the potential to be used long-term for routine vaccination in the future.

## 2. Results

### 2.1. Design and Preparation of a Trimeric SARS-CoV-2 Spike Antigen

To generate the SARS-CoV-2 S protein in the prefusion state, we designed a construct that incorporated a “GSAS” replacement at the furin cleavage site (RRAR) and two proline substitutions at residues 986 and 987 (K986P, V987P) in the sequence of the wild-type spike from the Wuhan strain, named S-2P. Additionally, to facilitate S protein trimerization and improve conformational homogeneity, another construct (S-Trimer) was designed by adding a trimerization domain, IZN4 [25], to the C-terminus of the S-2P construct. To confirm the function of these modifications, a commercialized S protein (Acro-S) was also included as a reference (Figure 1A). The plasmids encoding the S-Trimer (or S-2P) were transfected into ExpiCHO cells (ThermoFisher Scientific, Carlsbad, CA, USA) for transient expression for 10 days. Then, serum-free culture medium was harvested by centrifugation, followed by purification through Ni^+^-NTA affinity chromatography. The SDS–PAGE analysis verified that both S variants were mostly expressed and secreted into the culture medium (Appendix A). The size of the recombinant S variants was assessed by reducing SDS–PAGE and nonreducing native PAGE (Figure 1B). With the help of these modifications, the purified S-Trimer or S-2P was polymerized into a stable homogeneous trimer, which appeared in multiple high molecular weight forms under native PAGE analysis. Under SDS–PAGE analysis, the reduced forms of S-Trimer and S-2P were resolved at a molecular weight of approximately 170 kDa (Figure 1B). By using biolayer interferometry, we next investigated whether S variants in the prefusion form affected the kinetics of the interaction with hACE2 (Figure 1C). The S-Trimer demonstrated the highest hACE2 binding affinity (KD of 7.3 nM) among these Sprotein; this was over two-fold higher than that of Acro-S (KD of 16 nM) and ~four-fold higher than that of S-2P (KD of 30 nM). These results indicated that the S-Trimer protein folded properly, formed a soluble and stable trimer, and bound to hACE2 with relatively high affinity.

### 2.2. Vaccination of S-Trimer with SWE Elicits Robust Humoral Immunity in Mice

We evaluated the immunogenicity of the SARS-CoV-2 S-Trimer at various dose levels and through different administration routes in BALB/c mice. The mice were vaccinated intramuscularly (IM) or subcutaneously (SC) twice in 4-week intervals (Figure 2A) with S-Trimer (0.5 μg or 5 μg or 20 µg) either nonadjuvanted or with adjuvants, 250 μg of aluminum hydroxide (Alum), or SWE (1:1 ratio by volume). Serum from the vaccinated mice was collected on Days 28 and 42. Via SC injection, all the vaccine groups exhibited a detectable antibody response against S-Trimer after a single-dose vaccination, as revealed by ELISA (enzyme-linked immunosorbent assay). The total IgG endpoint titer (Log10) of the serum among the vaccine groups collected at Day 28 ranged from two to three and subsequently increased approximately one order for each group after boosting at Day 42. SC vaccination with the S-Trimer adjuvanted and SWE elevated the level of anti-S binding antibody (Figure 2B,C), ACE2 competitive titer (Figure 3A), and viral neutralization titer (Figure 3B) more than vaccination without the adjuvant or with the Alum. Although various doses of S-Trimer antigen were included in each group, the humoral antibody response induced by the SWE-adjuvanted (S-Trimer/SWE) groups did not show significant differences among various antigen doses. Instead, a dose-dependent effect was observed on the anti-S binding antibody titer of the nonadjuvated (S-Trimer) and Alum-adjuvanted (S-Trimer/Aum) groups, which showed no significant difference in antibody responses compared to each other at the corresponding antigen dose level. On the other hand, IM vaccination with non, Alum-, or SWE-adjuvanted S-Trimer with various doses of antigen induced antibody responses, which did not markedly appear to be dependent on the amount of antigen used. Higher levels of anti-S binding, ACE2 blocking, and neutralizing antibody titers were measured in both groups that received Alum- and SWE-adjuvanted S-Trimer. The boost effect of the second dose with adjuvants (on Day 28) was evident, with over 10-fold increases in the anti-S binding antibody titers at Day 42, which were maintained until 8 weeks and slightly decreased in the following weeks (Appendix A). The ELISA endpoint titer (Figure 2D,E) and ACE2 competitive titer (Figure 3C) in the sera from the SWE-adjuvanted groups were statistically higher than those from the Alum-adjuvanted groups at the corresponding antigen dose level. However, the neutralizing antibody levels of the SWE-adjuvanted groups were numerically higher than those of the Alum-adjuvanted groups (Figure 3D). Taken together, these results indicate that SWE adjuvantation induced humoral antibody responses against SARS-CoV-2 that were stronger than those induced by Alum and were further improved via IM injection.

### 2.3. SWE Adjuvanticity Is Associated with the Induction of Tfh Cells and GC B Cells

It is known that the magnitude and quality of antibody responses and humoral memory are mediated through GC B cells, which differentiate into long-lived memory B cells or plasma cells through Tfh regulation [26,27]. To determine whether SWE enhanced S-Trimer-specific IgG and neutralization titers through an increase in Tfh and GC B cells, BALB/c mice were intramuscularly immunized using a treatment regimen described previously (Figure 2A) with various doses of S-Trimer that was either nonadjuvanted or adjuvanted. Then, we quantified the CD4^+^CXCR5^+^ICOS^high^PD-1^high^ Tfh cells (Figure 4A) and CD19^+^B220^+^GL-7^+^CD95^+^ GC B cells (Figure 4B) by flow cytometry in the draining lymph nodes (dLNs) two weeks after boosting the mice. Compared with the PBS, nonadjuvanted, or Alum-adjuvanted groups, SWE adjuvating markedly increased the proportion of Tfh (Figure 4C) and GC B cells (Figure 4D), but the proportions were not significantly different between the nonadjuvanted and Alum-adjuvanted groups. To define the relative contribution of the induction of Tfh cells or GC B cells to the SWE-elicited neutralizing antibody response, the correlations between the proportion of Tfh cells (or GC B cells) and NT titers were assessed in individual immunized mice. A positive direct correlation (R^2^ = 0.3445) was observed between Tfh cells and NT titers (Figure 4E), and a stronger positive correlation (R^2^ = 0.6093) was presented between GC B cells and NT titers (Figure 4F). This evidence indicates that the increase in the Tfh cells and GC B cells by SWE adjuvating facilitates the formation of high-affinity class-switched antibodies, leading to higher levels of neutralizing antibodies against SARS-CoV-2.

### 2.4. SWE-Adjuvated S-Trimer Enhances the T-Cell Response

Cellular immune responses are important to mediate the production of high-quality antibodies and eliminate virus-infected host cells. To assess whether SWE-adjuvanted S-Trimer vaccination promotes the development of spike-specific T-cells, BALB/c mice were intramuscularly immunized with various doses of S-Trimer either nonadjuvanted or adjuvanted twice at a 4-week interval. The mice were sacrificed two weeks after the second vaccination, and then we restimulated splenocytes ex vivo with the S-2P ectodomain, a CD4 epitope (S444-458 KVGGNYNYLYRLFRK), or a CD8 epitope (S535-543, KNKCVNFNF) [28] for two days. The T-cell responses were evaluated by quantitating the number of IFN-γ-secreting cells by ELISpot assay. The frequency of IFN-γ secreting cells was detected below 10 spots/10^6^ cells in the splenocytes from individual mice without any stimulation (Figure 5A). The S-Trimer/SWE-immunized groups induced distinctly high frequencies of IFN-γ secreting cells (~150 spots/10^6^ cells) after stimulation with recombinant S-2P (Figure 5B). However, S-Trimer/SWE induced strong S535-543 targeting of CD8 (Figure 5D but limited S444-458 targeting of CD4 T-cell responses. The S-specific CD4 responses were significantly enhanced only in mice immunized with 20 µg of S-Trimer adjuvanted by SWE (Figure 5C). In contrast, S-Trimer/Alum vaccination induced a relatively low frequency of IFN-γ secreting cells, similar to the PBS and S-Trimer groups, despite inducing comparable neutralizing antibody responses in BALB/c mice. Overall, the S-Trimer/SWE groups appeared to induce a stronger cell-mediated immune response than that of the nonadjuvanted (S-Trimer) or S-Trimer/Alum groups.

To assess the profile of splenocytic S-specific CD4 T helper (Th) cells in BALB/c (Figure 6A) and C57BL/6 (Figure 6B) mice, we determined the secretion of Th1 cytokines (IFN-γ and IL-2) and Th2 cytokines (IL-4, IL-5, and IL-13) from stimulated splenocyte supernatants by ELISA. The results showed that there was a relatively low but detectable cytokine response in the nonadjuvanted (S-Trimer) and Alum-adjuvanted (S-Trimer/Alum) groups. Compared to the S-Trimer- or S-Trimer/Alum-immunized groups, Th1 and Th2 cytokine secretion from S-specific T-cells was obviously increased in S-Trimer/SWE-immunized BALB/c and C57BL/6 mice. Among these S-Trimer/SWE groups, the highest levels of IL-2 and IFN-γ cytokines were observed in both strains of mice that were immunized by the lowest dose (0.5 µg) of S-Trimer with SWE adjuvating. This lowest dose formulation with SWE also induced the highest levels of IL-4, IL-5, and IL-13 cytokines in C57BL/6 mice. In contrast, SWE adjuvating did not cause a significant difference in IL-5 or IL-13 production among BALB/c mice treated with various doses of S-Trimer, while the secretion of IL-4 increased with increasing amounts of S-Trimer by SWE adjuvating. Furthermore, vaccine-induced bias of the cytokine profile was determined in accordance with the ratio of IFN-γ/IL-4. Since Alum and SWE have been considered Th2-biased adjuvants that naturally induce a higher proportion of Th2 than Th1 cytokines, a relatively low ratio of IFN-g/IL-4 was observed in the S-Trimer/Alum and S-Trimer/SWE groups. Interestingly, SWE adjuvanted with the lowest dose (0.5 µg) of S-Trimer resulted in a significantly higher ratio of IFN-γ/IL-4, demonstrating a pro-Th1-cell-mediated immune response. The Th1-biased CD4 T-cell response appears to be associated with a decrease in the dose of the S-Trimer used when adding the SWE adjuvant.

### 2.5. Humoral Immune Responses and Protective Efficacy in Vaccinated Syrian Hamsters

We next evaluated the protective efficacy of the S-Trimer vaccines to prevent viral infections and associated diseases since SARS-CoV-2 showed limited binding to mouse ACE2, which has limited the use of inbred mice for research. Hamster ACE2 binds tightly to the SARS-CoV-2 S protein and mediates its entry [29]; therefore, hamsters are a competent infection model for studying the pathogenesis of SARS-CoV-2 infections [30]. In this study, Syrian hamsters were intramuscularly immunized twice at a 4-week interval and intranasally infected with SARS-CoV-2 on Day 45 (Figure 7A). Serum samples were collected on Day 42 and subjected to different methods to assess antibody responses. The levels of ELISA (Figure 7B), ACE2 blocking (Figure 7C), and live-virus neutralization titers (Figure 7D) measured in sera from the S-Trimer/SWE groups were significantly higher than those from the nonadjuvanted S-Trimer groups at each corresponding antigen dose level or the S-Trimer/Alum groups at the lowest antigen dose (0.5 μg).

In the challenge study, the hamsters were vaccinated with non, Alum-, or SWE-adjuvanted 0.5 μg of S-Trimer, infected with 200 TCID50 of SARS-CoV-2, and monitored for body weight loss for 6 days post-infection (dpi). The hamsters from each group were separately sacrificed on dpi 3 and dpi 6 to harvest lungs for histopathological analysis and measure virus titers in the lung. After infection, the hamsters in the SWE-adjuvanted S-Trimer groups were protected from body weight loss, while the hamsters in the PBS control group showed a gradual body weight loss of approximately 10% through 6 dpi (Figure 7E and Appendix A). The body weight of the hamsters in the S-Trimer and S-Trimer/Alum groups slightly decreased on dpi 2 to 4 and then recovered and experienced less weight loss than that of the PBS control group. At 3 days post-virus-infection, the viral titer in the lung indicated high viral replication in hamsters from the PBS control group as well as the nonadjuvated S-Trimer group. A significant reduction in the virus titers was observed in the lungs from the S-Trimer/Alum group, but two hamsters still had detectable virus. However, the viral load in the lungs from the S-Trimer/SWE group was completely suppressed below the detectable limit (Figure 7F and Appendix A).

A histopathology analysis at dpi 6 also revealed that different severities of inflammation and tissue damage were observed in the lungs from all the groups except for hamsters from the S-Trimer/SWE groups, which did not show any lung lesions (Figure 7G,H, Appendix A). Various parameters, as above, suggested that SWE adjuvating the S-Trimer vaccine can fully protect hamsters from SARS-CoV-2 infection.

### 2.6. SWE-Adjuvanted S-Trimer Induces Cross-Neutralizing Activity against VOCs

Here, we evaluated the neutralization ability of the serum samples collected from the hamsters vaccinated with various doses of S-Trimer with SWE adjuvant against the SARS-CoV-2 Beta (B.1.351) or Delta (B.1.617.2) variants compared to the neutralization of wild-type SARS-CoV-2 (Figure 8). The results indicated that there were minimal effects on low doses of S-Trimer (0.5 μg) with SWE-induced neutralization titers against the Beta variant (2.0-fold reduction) or Delta variant (1.08-fold reduction) compared with that of the WT; higher doses of S-Trimer (5 μg or 20 μg) with SWE showed statistically decreased neutralization titers against the Beta variant (4.36-fold or 4.0-fold reduction) or Delta (2.35-fold or 2.52-fold reduction) compared with that of the WT. These results suggested that increasing the amount of S-Trimer in the SWE adjuvant can elevate the neutralizing antibody titer against the WT of SARS-CoV-2, but this effect did not observed in the neutralization assay for the Beta or Delta variants.

## 3. Discussion

Different types of COVID-19 vaccines have been approved or used in humans with emergency authorization. Although many recombinant protein vaccines for COVID-19 have been investigated in clinical trials, limited vaccines (Covifenz (CoVLP), Abdala (CIGB 66), MVC-COV1901 NVX-CoV2373, Noora, and ZF2001 (ZIFIVAX)) have been approved for emergency use in different countries [31]. It is important to use adjuvants to generate Th1-biased immune responses. It has shown that MF59 formulated with HIV or influenza vaccine can increase antibody titers [32]. SWE is similar to the MF-59; we expect that the strong adjuvanted activity can reduce the doses of the antigens. In this report, we found that low-dose antigen in an emulsion-type adjuvant SWE can induce Th1-biased immune responses in the absence of CpG (TLR9 agonist). Our findings could reduce the costs of COVID-19 vaccines. Although the SWE did not have a product in the market, the similar formula MF59 has been used in humans for many years. We expect that a vaccine product containing SWE will be in the market in the near future.

Although many recombinant protein vaccines have been studied using different adjuvants, aluminum- or emulsion-type adjuvants alone have been investigated less. In this report, we designed a trimeric S of SARS-CoV-2 formulated with aluminum or SWE to evaluate the potential for COVID-19 vaccines. Surprisingly, the low dose of trimeric S with SWE generated protective immunity against SARS-CoV-2 infection in hamsters and induced Th1-biased immune responses (Figure 5 and Figure 7). NVX-CoV2373 is a nanoparticle protein vaccine that is adjuvanted with Matrix M to obtain balanced Th1/Th2 immune responses [14]. MVC-COV1901 and SCB-2019 are prefusion forms of trimeric S protein formulated with Alum/CpG that can induce Th1-biased immune responses [33,34]. They can induce protective immunity against virus infection. In contrast, recombinant protein formulated with gold nanoparticles can elicit anti-S IgG antibodies but fails to induce protective immune responses [35]. The extracellular domain of the spike protein was fused to a molecular clamp trimerization domain from glycoprotein 41 (amino acids 540–576 and 619–656) of human immunodeficiency virus 1 (HIV-1) to generate a prefusion-stabilized spike protein (Sclamp). The Sclamp in MF59 elicits high levels of neutralizing antibodies and protective immunity [10]. Unfortunately, the vaccine candidate was withdrawn due to a cross-reaction with HIV diagnoses. This result indicated that the trimerization domain needs to have low immunogenicity. Our trimeric S is trimerized with the low immunogenic sequence IZN4, which is derived from GCN4 [25]. Our design will be more practical in future applications in mass vaccination.

In addition to neutralizing antibody titers, T-cells also play an important role in controlling disease progression. We analyzed T-cell responses using whole S protein, CD4 epitope peptide, and CD8 epitope peptide. S-Trimer/SWE immunization elicited higher levels of both CD4+ and CD8+ T-cell responses than S-Trimer/Alum immunization (Figure 6). Our results are consistent with Sclamp in MF59 immunization, inducing higher levels of T-cell responses than Sclamp in alhygrogel immunization [10]. Although protein antigen immunization does not easily induce CD8+ T-cell responses, the selected adjuvant probably can enhance CD8+ T-cell responses. Viral-specific CD8+ T-cells may kill infected cells to control disease progression. The depletion of CD8+ T-cells in convalescent rhesus macaques partially abrogated the protective efficacy of natural immunity against rechallenge with SARS-CoV-2 [36]. The depletion of both CD4+ and CD8+ T-cells in mice led to an inability to clear the virus during primary infection [37]. These studies indicated that vaccine-induced T-cell responses play an important role in controlling the severity of COVID-19.

Due to the rapid evolution of SARS-CoV-2, COVID-19 vaccines are expected to neutralize recent VOC infections. Growing evidence shows that VOCs have a negative impact on the efficacy of the current vaccines, in which the antigens have been designed based on the sequence of spikes from the ancestral Wuhan strain [38]. Immune evasion by emerging variants was caused by several key mutations in the spike protein, especially in the RBD. For example, a Beta strain (B.1.351) that encodes the K417N/E484K/N501Y mutations in RBD abolished the neutralizing activity in the plasma from individuals vaccinated with BNT162b2 (Pzifer/BioNTech) or mRNA-1273 (Moderna), who exhibited reductions of 10.3- to 14-fold or 3.8- to 8.4-fold, respectively, when compared to that of the B.1 (D614G) strain [39,40,41,42,43]. Moreover, the mRNA vaccine elicited neutralizing activity against the Delta strain (B.1.617.2), including L452R and T478K mutations in the RBD, with a 2.38-fold reduction for BNT [44] and a 3.3-fold reduction for Moderna [43]. Our data showed that S-Trimer/SWE can induce cross-reactive neutralizing antibody titers against SARS-CoV-2 VOCs. Even though a minor reduction in the neutralization abilities against the Beta and Delta variants was observed, serum from S-Trimer/SWE-vaccinated hamsters contains neutralizing antibody levels on average GMT 141 (Beta) or 248 (Delta), which are higher than the neutralizing antibody level (GMT 85) against the WT induced by 0.5 μg S-Trimer/Alum vaccination that offers protection from viral infection in hamsters. Taken together, these data demonstrate that reduced but still protective humoral immunity may be retained against the Beta and Delta variants following S-Trimer/SWE vaccination.

The shortage of the COVID-19 vaccines and ultralow temperature storage limited the global vaccine distribution. The subunit vaccine is safe and stored in the freezer. Although the vaccine manufacturing capacity is limited in the present, the reduction of the antigen doses can increase the supply of the vaccine. The current subunit COVID-19 vaccines contained 5 to 15 μg antigen [17,45]. If the antigen amount can be reduced to 1/10, the vaccine can increase 10-fold in supply. These points indicate that adjuvant is critical in the vaccine supply and resolves the shortage of the vaccines. The antigen design and selection of adjuvants are important for the emerging infectious diseases or routing influenza vaccines vaccination.

In summary, our report demonstrated that a low dose of recombinant trimeric S formulated with WSE can induce both humoral and cellular immune responses to protect animals from SARS-CoV-2 infection. Our findings can be applied to other infectious diseases and reduce the cost of vaccines.

## 4. Materials and Methods

### 4.1. Production of SARS-CoV-2 S-Trimer

To produce secreted SARS-CoV-2 S proteins with a prefusion form, a DNA fragment encoding the Wuhan strain S protein was designed to contain a nonfunctional furin cleavage site (R682G, R683S, R685S) and two stabilizing prolines (K986P, V987P) in the hinge loop. Additionally, the transmembrane domain and the C-terminal intracellular tail were removed (S-2P) or replaced by a trimerization domain IZN4 (S-Trimer), followed by histidine (8-mer) at the carboxy terminus for purification. Both S variant DNA constructs were codon-optimized for Homo sapiens, synthesized, and cloned into the pcDNA3.1(+) plasmid vector by GenScript (Piscataway, NJ, USA). The production of the S-Trimer was compiled using the ExpiCHO™ Expression System Kit (ThermoFisher Scientific, Carlsbad, CA, USA). Briefly, the S-Trimer (or S-2P) was transiently expressed by ExpiCHO cells with serum-free medium according to the manufacturer’s instructions. The culture medium containing S-Trimer was centrifuged at 15,000 rpm for 30 min at 4 °C; later, the supernatant was dialyzed to equilibration buffer (50 mM Tris-HCl, 150 mM NaCl, and 20 mM imidazole; pH 8.9). S-Trimer was purified through an equilibrated Ni^2+^-NTA agarose column (GE). Finally, the S-Trimer was eluted by equilibration buffer with 0.5 M imidazole and dialyzed against buffer without imidazole (20 mM sodium phosphate; pH = 8.0).

### 4.2. Animal Immunization

BALB/c mice and C57BL/6 and Syrian hamsters were obtained from the National Laboratory Animal Breeding and Research Center (Taipei, Taiwan). Mice or hamsters were used between 6 and 12 weeks of age. Anesthetized animals were vaccinated with S-Trimer in aluminum hydroxide (InvivoGen, San Diego, CA, USA) or an equal volume of SWE solution (SEPPIC, La Garenne-Colombes, France) at 2- or 4-week intervals. Blood samples of mice or hamsters were collected by tail vein or submandibular blood sampling. All animals were housed at the Animal Center of the National Health Research Institutes (NHRI) and maintained in accordance with institutional animal care protocols. All animal experimental protocols were approved by the Institutional Animal Care and Use Committee (IACUC) of the NHRI (Protocol No: NHRI-IACUC-109077-A, approval date: 1 May 2020).

### 4.3. Immunoassay

The SARS-CoV-2 S binding antibody titers in serum samples collected from vaccinated animals were evaluated by ELISA (enzyme-linked immunosorbent assay). Purified S-IZN4 in PBS (0.2 μg/100 μL/well) was incubated in 96-well microplates (Corning, Glendale, Arizona, USA) at 4 °C overnight. After the coating solution was removed, the plates were blocked with 3% BSA in PBS at RT for 2 h and washed three times with 0.1% PBS-T. Next, serum samples at 1:30 were threefold serially diluted with 1% BSA in PBS and applied to wells (100 μL). After incubating at RT for 2 h, the plates were washed three times with 0.1% PBS-T, followed by incubation with species-specific secondary antibodies, including goat anti-mouse IgG HRP (Cappel, Solon, OH, USA) and goat anti-hamster IgG-HRP (Arigo Biolaboratorie, Hsinchu City, Taiwan, ROC), at RT for 1 h. The plates were then washed three times with 0.1% PBS-T, and signals were developed using SureBlue TMB substrate (Seracare, Milford, MA, USA). The colorimetric reaction was stopped by adding 2 M H_2_SO_4_. The optical density (OD) was measured at 450 nm by an ELISA reader (BioTek, Winooski, VT, USA).

### 4.4. ACE2 Competition ELISA

Purified S-2P protein was coated into the wells of 96-well microplates (0.2 μg/100 μL/well) in PBS, incubated at 4 °C overnight, and blocked with 3% BSA in PBS at 37 °C for 2 h. After being washed with 0.1% PBS-T, the serially diluted serum samples and a reference anti-SARS-CoV-2 neutralizing antibody (50 μL/well) were incubated in the microplates for 1 h, followed by mixing with 50 μL of biotinylated human ACE2 (0.2 μg/mL) in each well and incubating for another 1 h. After being washed with 0.1% PBS-T, the microplates were incubated with a 1:5000 dilution of streptavidin-HRP (BD Biosciences, San Jose, CA, USA) for 30 min. Wells were washed with 0.1% PBS-T again, and the TMB substrate (Seracare, Milford, MA, USA) was added for signal development. Then, the colorimetric reaction was stopped by adding 2 M H_2_SO_4,_ and the OD450 nm was measured by a microplate reader.

### 4.5. Neutralizing Antibody Analysis

Neutralizing antibody titers against live SARS-CoV-2 virus were evaluated by the TCID50 (median tissue culture infectious dose) assay. Live viruses (hCoV-19/Taiwan/4/2020 (D614), hCoV-19/Taiwan/1013/2021, or hCoV-19/Taiwan/1144/2021) were generated as previously described [46]. All serum samples were inactivated at 56 °C for 30 min and serially diluted in M199 medium in 2-fold dilutions starting from 1:20. The diluted serum was mixed with an equal volume of solution containing 200 TCID50 of SARS-CoV-2 virus with a volume of 200 μL per well. After 2 h of incubation at 37 °C, the virus–serum mixtures were transferred to a 96-well plate covered by a monolayer of Vero cells and cultured at 37 °C. Quadruplicates were prepared for each serum dilution. Cytopathic effect (CPE) characteristics in wells were recorded under microscopes after 4–5 days, and the neutralization titer was determined as the reciprocal of serum dilution required for preventing infection of 50% of quadruplicate inoculations. Neutralization titers below the starting dilution of 1:20 were assigned a value of 10 for calculation purposes.

### 4.6. Flow Cytometric Analysis

The draining lymph nodes (LNs) were homogenized in RPMI-1640 and filtered through a 70-μm nylon strainer (BD Biosciences, San Jose, CA, USA). LN single-cell suspensions were stained in FACS buffer (PBS, 2% FBS, 1 mm EDTA, and 0.1% sodium azide) and incubated with anti-CD16/CD32 antibody (BD Biosciences, San Jose, CA, USA) for 30 min on ice to block nonspecific binding on Fc receptors. Subsequently, the cells were stained with a cocktail of monoclonal antibodies for 1 h on ice. To stain GC B cells and TFH cells, lymphocytes were incubated with anti-CD19-PE, anti-B220-PE-Cy7, anti-GL7-FITC, and anti-CD95-APC Abs or with anti-CD4-BV421, anti-CXCR5-PE, anti-ICOS-APC, and anti-PD-1-PerCPCy5.5 Abs.

### 4.7. Cytokine Assay

Splenocytic cytokine production was assessed using cytokine ELISA. A spleen was isolated from individual mice in each group at two weeks after the second vaccination. Splenocytes were prepared by centrifugation of mesh-homogenized spleens in RPMI-1640 medium and suspended in LCM with 10% fetal bovine serum. The obtained splenocytes were cultured with recombinant SARS-CoV-2 S (10 μg/mL) in 24-well plates (5 × 10^5^ cells/well) at 37 °C under 5% CO_2_. After 3 days, the supernatant was collected to quantify the production levels of Th1 cytokines (IFN-γ and IL-2) and Th2 cytokines (IL-13, IL-5, and IL-4) by using separate cytokine ELISA kits (ThermoFisher Scientific, Carlsbad, CA, USA) as described by the manufacturer’s instructions.

### 4.8. ELISPOT Assay

The number of IFN-γ-secreting cells in the spleen was evaluated using a mouse IFN-γ ELISPOT assay kit (BD Biosciences, San Jose, CA, USA) according to the manufacturer’s instructions. IFN-γ capture antibody (1:200) was coated on 96-well plates with PVDF membranes (Merk & Millipore, Burlington, MA, USA) by incubation at 4 °C overnight. After washing with PBS, the plates were blocked with complete RPMI-1640 for 2 h. The splenocytes (5 × 10^5^ cells/well) were seeded into the plates in duplicate wells with 10 μg/mL recombinant S-2P or synthetic peptides (S444-458 KVGGNYNYLYRLFRK and S535-543 KNKCVNFNFNF), which have been reported as CD4 and CD8 epitopes in BALB/c [28]. After the splenocytes were restimulated in 5% CO2 at 37 °C for 2 days, the cells were removed from the plates by washing three times with PBST. Then, the plates were incubated with a biotinylated detection antibody (1:250) at 37 °C for 2 h. After repeating the above washing steps, avidinHRPcomplex reagent (1:100) was added to the plate and incubated at room temperature for 45 min. The plates were washed three times with PBST and then three times with PBS alone. A staining solution (3-amine-9-ethylcarbazole, Merck/MilliporeSigma, Burlington, MA, USA) was added to the wells to develop the spots. After 30 min, the plates were placed under tap water to stop the reaction. The spots were determined by an ELISPOT reader (Cellular Technology Ltd., Shaker Heights, OH, USA).

### 4.9. Animal Challenge

The animal model for SARS-CoV-2 infection in Syrian hamsters was conducted as previously described [46]. Syrian hamsters were intramuscularly immunized with either PBS as a negative control, 0.5 μg S-Trimer, or 0.5 μg S-Trimer mixed with adjuvant (Alum or SWE). Each injection volume of PBS control and mixed vaccines (antigen + adjuvant) was 100 μL per IM dose. Four groups of hamsters (*n* = 8/group) were immunized twice (on Days 0 and 28) and then infected intranasally with a dose of 1 × 10^5^ TCID50 SARS-CoV-2 virus on Day 45. The SARS-CoV-2 strain (hCoV-19/Taiwan/4/2020 (D614G)) used was obtained from the Centers for Disease Control (CDC, Taipei city, Taiwan, ROC) in Taiwan. After viral infection, four hamsters in each group were sacrificed at Day 3 for viral load quantification. The body weights of the other four hamsters in each group were recorded daily until Day 6 before being sacrificed. To determine the viral load in the lung, left lung tissues were homogenized in 2 mL of PBS using a gentle-MACS^®^ Dissociator (Miltenyi Biotec, Bergisch Gladbach, North Rhine-Westphalia, Germany). After centrifugation at 600× *g* for 5 min, the clarified supernatant was harvested for live virus titration (TCID50 assay) and viral RNA quantification.

### 4.10. H&E Staining

To evaluate lung histopathology by H&E staining, lung tissues from hamsters were fixed in 10% formalin and paraffin embedded. Sections were prepared and stained with hematoxylin and eosin (H&E). Lung pathology, including overall lesion extent, pneumocyte hyperplasia, and inflammatory infiltrates, was assessed by a clinical pathologist at the core pathology facility of the National Health Research Institutes (Miaoli County, Taiwan). Scores were determined based on the percentage of the inflammation area for each section of total lobes collected from each animal in each group by using the following scoring system: 0 = no pathological change; 1 = infiltration area ≤10%; 2 = infiltration area 10%; 3 = infiltration area ≥50%. An additional point was added when pulmonary edema and/or alveolar hemorrhage was observed. The total score for all the lobes in the image is shown for individual animals.

### 4.11. Statistical Analysis

Statistical data arrangement was generated using GraphPad Prism software. Two-tailed Mann–Whitney tests were used to compare the value of antibody titer measured by different methods, the number of IFN-γ-secreting cells, and cytokine secretion level between two experimental vaccination groups. The viral load in lung and pathological severity of lung tissue among multiple groups were performed using Kruskal–Wallis ANOVA with Dunn’s multiple comparison tests. Comparisons of hamster body weight change (%) were made by two-way ANOVA with multiple comparison tests at different time points; *p* values < 0.05 were considered significant; ns: not significant.

## Figures and Tables

**Figure 1 ijms-23-04902-f001:**
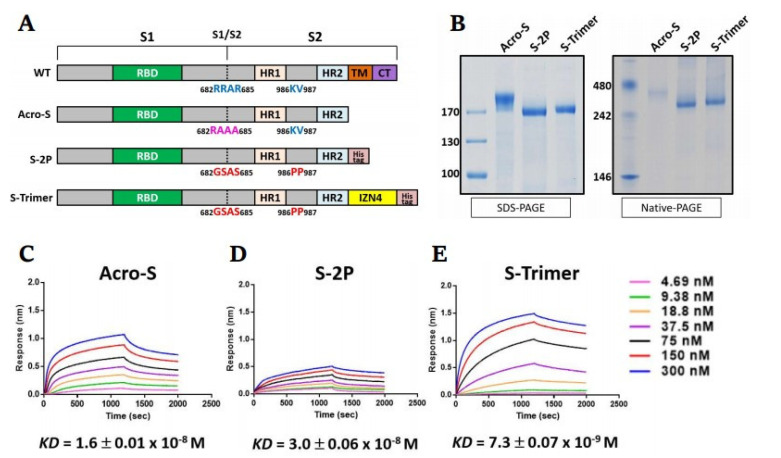
Design and characterization of recombinant SARS-CoV-2 spike variants. (**A**) Schematic design of SARS-CoV-2 S variants construct. Wild-type (WT) full length of SARS-CoV-2 S protein contents, RRAR residues at S1/S2, and a furin cleavage site. Acro-S has the S ectodomain without the furin cleavage site, which was substituted by ‘‘RAAA” at S1/S2. The recombinant SARS-CoV-2 S-2P protein encodes S ectodomain with a ‘‘GSAS” substitution at the furin cleavage site and two proline substitutions at residues 986 and 987 to retain in the prefusion conformation; S-Trimer was designed as S-2P fused with IZN4 trimerization domain at the C-terminal. (**B**) Reducing SDS–PAGE and native PAGE analysis of purified S-2P and S-Trimer. Molecular weight standards are indicated at the left in kDa. Kinetic profiles of (**C**) Acro-S, (**D**) S-2P, and (**E**) S-Trimer binding to human ACE2 measured by ForteBio BioLayer interferometry (BLI). The data are representative of at least three independent experiments.

**Figure 2 ijms-23-04902-f002:**
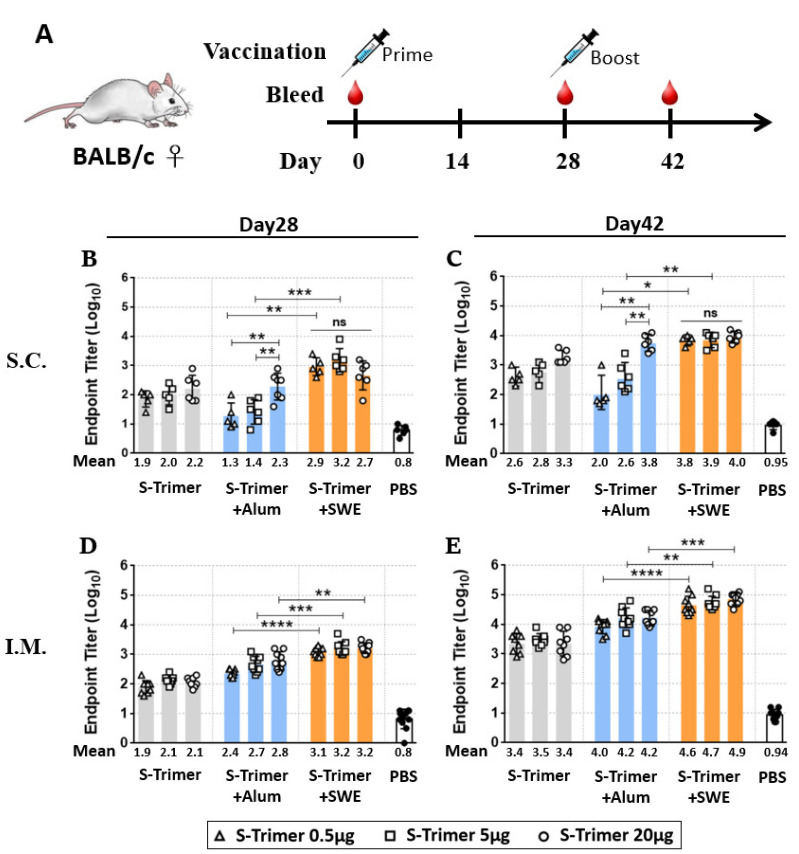
Detection of S-Trimer-specific antibodies in serum of vaccinated BALB/c mice. (**A**) Schematic showing the schedule for vaccination and serum collection. BALB/c mice were immunized with various doses of S-Trimer (0.5 μg, 5 μg, or 20 μg) that was nonadjuvanted (gray) or adjuvanted with 250 μg Alum (blue) or SWE (orange) twice on Day 0 and 28. The PBS-injected group was the blank control. Serum was collected on Day 0, 28, and 42. Using ELISA to access IgG titers against S-Trimer in serum, which collected after the first or second dose vaccination via (**B**,**C**) subcutaneous (SC) injection, *n* = 5–6/group, or (**D**,**E**) intramuscular (IM) injection, *n* = 8–9/group. Each symbol represents the endpoint titer from an individual mouse; each bar represents the geometric mean ± 95% confidence interval (CI) of IgG endpoint titer from the group. Statistically significant differences compared within adjuvanted vaccine groups by two-tailed Mann–Whitney test. * *p* < 0.05, ** *p* < 0.008, *** *p* < 0.0005, **** *p* < 0.0001; ns: not significant.

**Figure 3 ijms-23-04902-f003:**
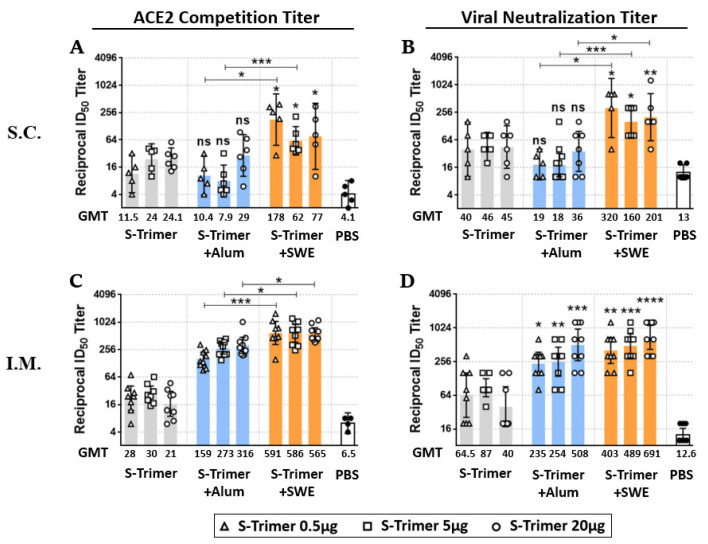
Analysis of ACE2 competitive and neutralizing antibodies in serum of vaccinated BALB/c mice. Inhibition of ACE2 binding to SARS-CoV-2 S-2P and live SARS-CoV-2 virus neutralization were accessed in serum (Day 42) collected from BALB/c mice after the second vaccination dose via (**A**,**B**) subcutaneous injection, *n* = 5–6/group, or (**C**,**D**) intramuscular injection, *n* = 8–9/group. Each symbol represents the reciprocal 50% inhibitory dilution ID_50_ titer from an individual mouse; each bar represents the geometric mean ± 95% CI ID_50_ titer from the group. Statistically significant differences were compared with the nonadjuvant vaccine group or adjuvanted vaccine group by two-tailed Mann–Whitney test. * *p* < 0.05, ** *p* < 0.008, *** *p* < 0.0005, **** *p* < 0.0001; ns: not significant.

**Figure 4 ijms-23-04902-f004:**
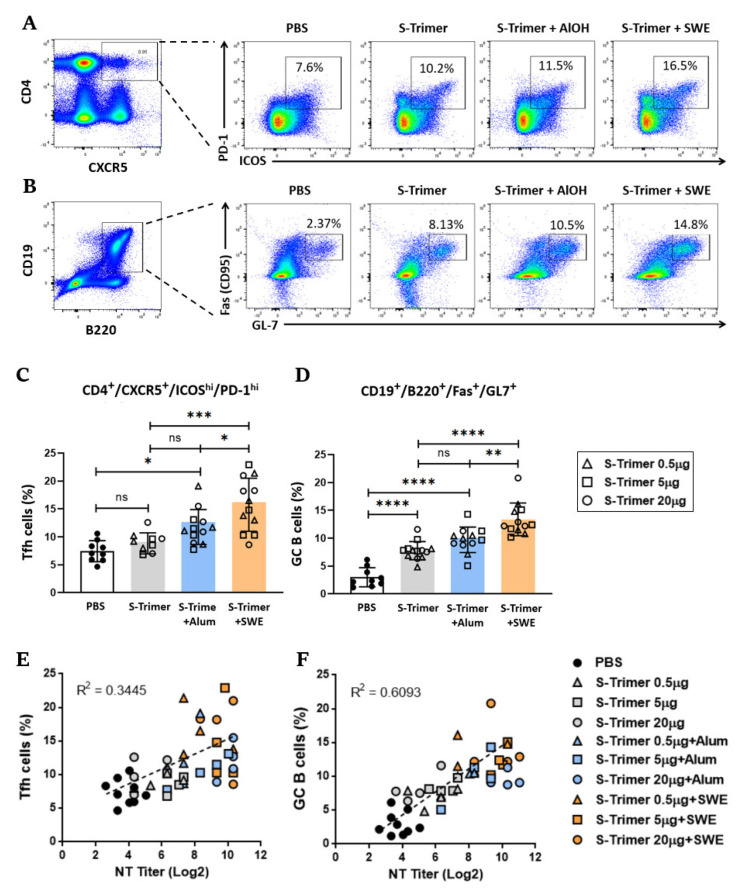
Increased number of Tfh and GC B cells in dLNs from vaccinated BALB/c mice. BALB/c mice (*n* = 3–4/group) were administered PBS as a control or 0.5 μg or 5 μg or 20 μg of S-Trimer protein with or without adjuvant (Alum or SWE) via intramuscular (IM) injection twice on Day 0 and 28. The dLNs were analyzed by flow cytometry at two weeks after the second vaccination. Representative dot plots of lymphocytes from immunized mice and gated on (**A**) CD4**^+^**CXCR5**^+^**ICOS**^high^**PD1**^high^** Tfh cells and (**B**) CD19**^+^**B220**^+^**GL-7**^+^** Fas**^+^** (CD95) GC B cells and percentage of (**C**) Tfh cells and (**D**) GC B cells in dLNs. Correlations between neutralization titer and percent of (**E**) Tfh cells or (**F**) GC B cells for each mouse. Symbols represent individual mice; bars indicate mean ± SD percentage of cells. Statistical analysis of comparisons among groups was made by two-tailed Mann–Whitney tests. * *p* < 0.05, ** *p* < 0.008, *** *p* < 0.0005, **** *p* < 0.0001; ns: not significant. The Spearman rank test was used to perform correlation analysis.

**Figure 5 ijms-23-04902-f005:**
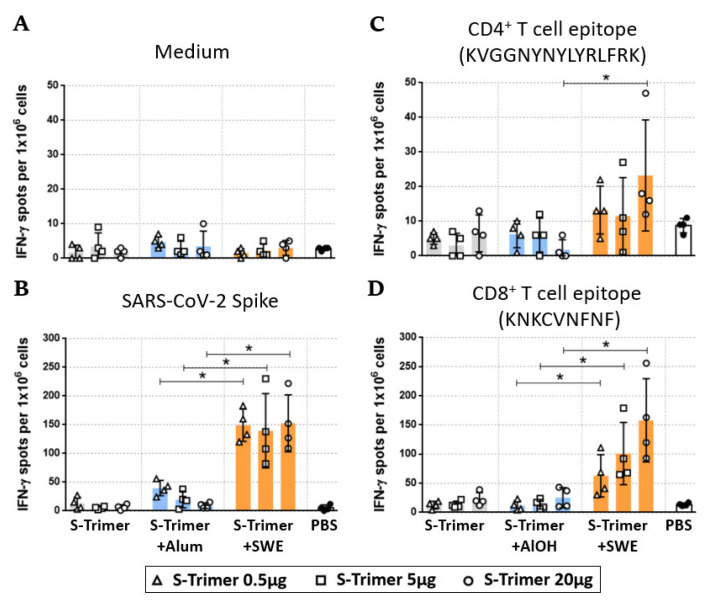
Spike-specific T-cell responses to adjuvanted S-Trimer immunization in BALB/c. Spleens were collected from the PBS control or immunized BALB/c mice (*n* = 4/group) two weeks after the second vaccination. Splenocytes from each individual were (**A**) cultured in medium as control or stimulated with (**B**) SARS-CoV-2 spike ectodomain, (**C**) CD4^+^ T-cell epitope, or (**D**) CD8^+^ T-cell epitope. The number of IFN-γ-secreting cells was evaluated by ELISpot assay. Each symbol represents individual mice; each bar represents the mean ± SD of each group for ELISpot assay. Statistically significant differences were compared by two-tailed Mann–Whitney test. * *p* < 0.05 was considered significant; ns: not significant.

**Figure 6 ijms-23-04902-f006:**
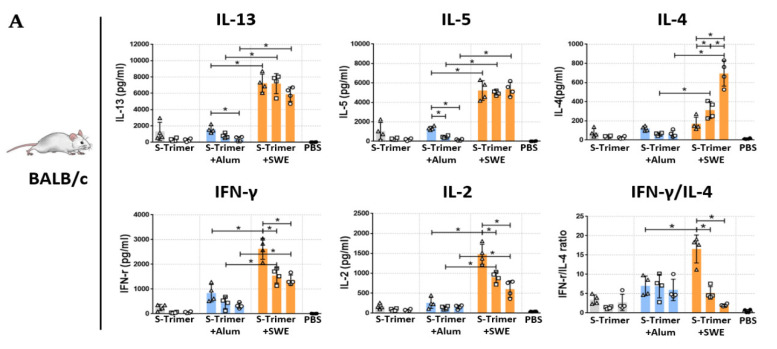
Cytokine production by spike-stimulated splenocytes from vaccinated mice BALB/c (**A**) and C57BL/6 (**B**) mice (*n* = 3–4/group) were administered PBS as a control or 0.5 μg or 5 μg or 20 μg of S-Trimer protein with or without adjuvant (Alum or SWE) via intramuscular (IM) injection twice on Day 0 and 28. Splenocytes were collected at two weeks from the second vaccination and incubated with SARS-CoV-2 S ectodomain to restimulate cytokine secretion. The secretion level of Th2 cytokine (IL-13, IL-5, and IL-4), and Th1 cytokine (IFN-γand IL-2) was evaluated by indirect ELISA. The Th1/Th2 ratio was calculated from each producing amount of IFN-γ and IL-4. Each symbol represents the cytokine value from an individual mouse; each group of cytokines’ value was presented as the mean ± SD. Statistical analysis of all comparisons was made by two-tailed Mann–Whitney tests. * *p* < 0.05 was considered significant.

**Figure 7 ijms-23-04902-f007:**
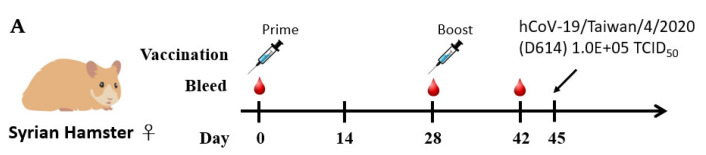
Protective efficacy of S-Trimer immunization against SARS-CoV-2 infection in Syrian hamsters. (**A**) Schematic showing the schedule of S-Trimer vaccination, SARS-CoV-2 infection, and blood collection for the hamster study. Syrian hamsters (*n* = 6–10/group) were administered with nonadjuvanted or adjuvanted S-Trimer 0.5 μg twice via intramuscular routes on Day 0 and 28. PBS-injected group as blank control. Serum was collected by gingival blood sampling on Day 0, 28, and 42. Syrian hamsters were intranasally infected on Day 45 with 1 × 10^5^ TCID50 live SARS-CoV-2 virus. The humoral immune responses were evaluated based on (**B**) anti S-Trimer Endpoint IgG titers, (**C**) ACE2 competition ELISA titers, and (**D**) wild-type SARS-CoV-2 virus neutralization titers. (**E**) Body weight change (%) of the hamsters was recorded every day after SARS-CoV-2 infection. (**F**) Virus titers in the lungs of SARS-CoV-2-infected hamsters at 3 days post-infection (dpi3) were determined by TCID50 assay. (**G**) Representative histopathology images of H&E staining from the lung sections of infected hamsters at 6 days post-infection (dpi6), in which purple indicates areas of inflammation. (**H**) Pathological severity of lung lesions was evaluated by the percentage of inflammation area of each section. Symbols represent individual animals; bars indicate geometric mean ± 95%CI of endpoint titers and ID_50_ titers for each group; horizontal lines indicate mean ± SD. For statistical analysis of antibody titers, virus titers, and pathological severity, significant differences compared between groups by two-tailed Mann–Whitney test. The comparisons of body weight change (%) were made by two-way ANOVA with multiple comparison tests. * *p* < 0.05, ** *p* < 0.008, *** *p* < 0.0005, **** *p* < 0.0001, ns: not significant.

**Figure 8 ijms-23-04902-f008:**
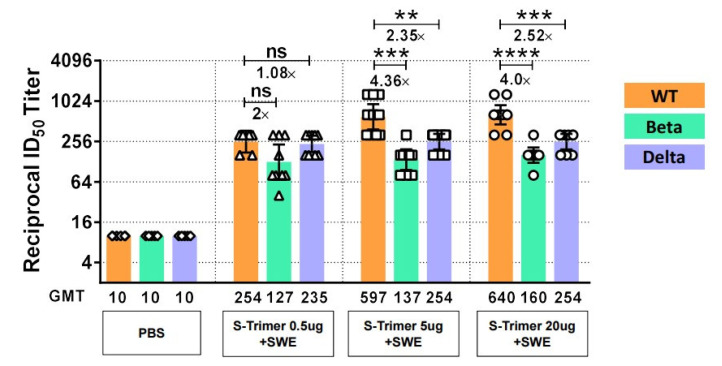
Cross-neutralization of SARS-CoV-2 variants by serum from S-Trimer with SWE vaccinated hamsters. Syrian hamsters (*n* = 6–10/group) were administered PBS as a control or 0.5 μg or 5 μg or 20 μg of S-Trimer protein (wild-type) with SWE adjuvant via intramuscular injection twice on Day 0 and 28. Cross-neutralizing antibodies against SARS-CoV-2 variants, B1.1351 (Beta) and B1.1617 (Delta), were assessed by TCID50 assay at two weeks after the second vaccination. Symbols represent the reciprocal 50% inhibitory dilution ID_50_ titer from an individual animal; bars represent the geometric mean ± 95% CI ID_50_ titer. Statistically significant differences were compared by two-tailed Mann–Whitney test. ** *p* < 0.008, *** *p* < 0.0005, **** *p* < 0.0001; ns: not significant.

## Data Availability

Not applicable.

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
