# Peer review of "Low-Dose SARS-CoV-2 S-Trimer with an Emulsion Adjuvant Induced Th1-Biased Protective Immunity"

_ijms, 2022, doi:10.3390/ijms23094902_

Round 1

Reviewer 1 Report

TITLE:  Low-dose SARS-CoV-2 S-Trimer with an emulsion adjuvant induced Th1-biased protective immunity

The aim of the present investigation was to assess the administration route, adjuvant formulation, and antigen dose of the S-Trimer vaccine.

GENERAL COMMENTS

The article is in-line with the journal topic, but some flaws should be improved.  The investigation is interesting, and the present paper is recommended for publication to the present journal after major revision.

Abstract

  1. The study background should be introduced by a single statement period.
  2. The materials and methods should be anticipated in this section.

Introduction

Line 39-54. This section is too generic and non-useful. The epidemiological part should be drastically reduced.

The viral vector transmission  mechanisms and barriers device efficacy should be debated in this section (PMID: 33670983; PMID: 32605056).

The aim of the study and the null-hypothesis should be discussed at the end of the section.

Materials and methods

The rationale of the animal study model is not completely clear. Why did you used two biologically different rodents models such as mice and Syrian hamsters?

Results

I noted that you evaluated both of female rodents’ species. Do you think that the gender and hormonal fluctuations could influence the study evidence?

Discussion

The limits of the present investigation should be added in this section.

The null-hypothesis should be discussed in this part of the manuscript.

Author Response

TITLE:  Low-dose SARS-CoV-2 S-Trimer with an emulsion adjuvant induced Th1-biased protective immunity

The aim of the present investigation was to assess the administration route, adjuvant formulation, and antigen dose of the S-Trimer vaccine.

GENERAL COMMENTS

The article is in-line with the journal topic, but some flaws should be improved.  The investigation is interesting, and the present paper is recommended for publication to the present journal after major revision.

Abstract

  1. The study background should be introduced by a single statement period.

Response:

We thank for the comments. The abstract has been edited by English editor (native speaker).

  1. The materials and methods should be anticipated in this section.

Response:

We thank for the kind reminder. The abstract has been revised and the materials and methods were added in this section. (P.1 line 19-36)

Introduction

  1. Line 39-54. This section is too generic and non-useful. The epidemiological part should be drastically reduced.

Response:

We appreciate the constructive suggestion. The first paragraph (P.1 line 39) has been rewritten to focus on the background of this study.

  1. The viral vector transmission mechanisms and barriers device efficacy should be debated in this section (PMID: 33670983; PMID: 32605056).

Response:

We appreciate the suggestion. However, the manuscript focuses on the potential of vaccines for routing vaccination. These two papers (PMID: 33670983; PMID: 32605056) are important but not highly related to this manuscript.

  1. The aim of the study and the null-hypothesis should be discussed at the end of the section.

Response:

We appreciate this thoughtful advice. We have made some modification to clarify the purpose and the null-hypothesis on the last paragraph (P.1 line 96-104) of introduction in the revised manuscript.

P.2 line 87 (highlighted in red)

In order to generate cost-effective, long-lasting and broad immune responses against evolving SARS-CoV-2 variants, this study is to assess the protective effects of antigen doses and adjuvants formulated with S-Trimer vaccine. The cross-neutralizing antibodies against SARS-CoV-2 Beta and Delta strains from different lineages were analyzed. As COVID-19 gradually goes from being an acute to endemic disease, such as seasonal influenza, the high safety subunit vaccine has the potential to be used long-term for routine vaccination in the future.

Materials and methods

  1. The rationale of the animal study model is not completely clear. Why did you used two biologically different rodents models such as mice and Syrian hamsters?

Response:

We apologize for the lack of clarity in our methods. The rationale of including Syrian hamster in animal study has been further described in the revised manuscript (P.16 line 246-250). Because SARS-CoV-2 showed good binding to human ACE2 but limited binding to mouse ACE2, which has limited the use of inbred mice for research. However, Syrian hamsters were originally determined to be susceptible to SARS-CoV [1]. Since hamster ACE2 binds tightly to the SARS-CoV-2 S protein and mediates its entry [2], therefore, hamsters are a competent infection model for studying the pathogenesis of SARS-CoV- 2 infections [3]. After infection, Syrian hamsters are able to allow SARS-CoV-2 replication within the lower respiratory tract with pathological symptoms as human COVID-19 pneumonia, including rapid breathing, lethargy, ruffled fur, and weight loss at 2 dpi [4].

P.16 line 246-250 (highlighted in red)

We next evaluated the protective efficacy of the S-Trimer vaccines to prevent viral infections and associated diseases. Since SARS-CoV-2 showed limited binding to mouse ACE2, which has limited the use of inbred mice for research. Hamster ACE2 binds tightly to the SARS-CoV-2 S protein and mediates its entry [33], therefore, hamsters are a competent infection model for studying the pathogenesis of SARS-CoV- 2 infections [34]. In this study, Syrian hamsters were intramuscularly immunized twice at a 4-week interval and intranasally infected with SARS-CoV-2 on Day 45 (Figure 7A).

Results

  1. I noted that you evaluated both of female rodents’ species. Do you think that the gender and hormonal fluctuations could influence the study evidence?

Response:

Biological differences between the sexes are likely to contribute to differences in the outcome of vaccination between the sexes. Immunological, hormonal, genetic and microbiota differences between males and females may also affect the outcome of vaccination [5]. Typically, females develop higher antibody responses following vaccination than males. These differences are observed in response to diverse vaccines, including the bacillus Calmette-Guerin vaccine, the measles, mumps and rubella vaccine, the yellow fever virus vaccine and influenza vaccines [6], as well as SARS-CoV-2 BNT162b2 mRNA vaccine [7-8]. In fact, we did not find a big difference in immune responses between the sexes from a pilot animal study. Besides, males usually injured in fights may cause bacterial infection or affect their diet and weight, therefore we routinely use female as vaccine recipients to minimize experimental errors.

Discussion

  1. The limits of the present investigation should be added in this section.

Response:  Thanks for the comments. The limitation of presented investigation has been added in the discussion section (the first paragraph).

  1. The null-hypothesis should be discussed in this part of the manuscript.

Response: Thanks for the comments. The null-hypothesis has been discussed in the discussion section (the first paragraph).

Reviewer 2 Report

Dear Authors,

I have read with interest the article titled Low-dose SARS-CoV-2 S-Trimer with an emulsion adjuvant induced Th1-biased protective immunity.
The article undoubtedly has considerable strengths: it is well-centered and thorough in pursuing its objective, and relies on sound methodology, as far as I could determine.
Another strength the article can boast lies in its relevance. Offering a broad array of data and findings buttressing the value of SWE as an adjuvant to enhance the immunogenicity of the S-Trimer vaccine certainly has valuable clinical and therapeutic potential, but the authors have been focused too closely (or solely) on their stated objective, thus completely neglecting a discussion necessary to contextualize the relevance of their findings. The article ends with 4.11. Statistical analysis, which is totally inappropriate in my view. Please lay out a cogent, comprehensive and thorough new section defining the prospective applications and utilization options which the data you have elucidated could pave the way for. You should make an effort to put such discoveries into perspective and place them within the current pandemic scenario, expounding upon their meaningfulness in an exhaustive fashion.

Although the article is quite well-written, I would have it proofread by a native speaker of English, in order to improve flow and readability overall, in addition to some minor grammar/vocabulary flaws.

This is a remarkably well-done research article and praiseworthy effort overall, hence I will be looking forward to a newly improved version.

Author Response

Dear Authors,

I have read with interest the article titled Low-dose SARS-CoV-2 S-Trimer with an emulsion adjuvant induced Th1-biased protective immunity.
The article undoubtedly has considerable strengths: it is well-centered and thorough in pursuing its objective, and relies on sound methodology, as far as I could determine.
Another strength the article can boast lies in its relevance. Offering a broad array of data and findings buttressing the value of SWE as an adjuvant to enhance the immunogenicity of the S-Trimer vaccine certainly has valuable clinical and therapeutic potential, but the authors have been focused too closely (or solely) on their stated objective, thus completely neglecting a discussion necessary to contextualize the relevance of their findings. The article ends with 4.11. Statistical analysis, which is totally inappropriate in my view. Please lay out a cogent, comprehensive and thorough new section defining the prospective applications and utilization options which the data you have elucidated could pave the way for. You should make an effort to put such discoveries into perspective and place them within the current pandemic scenario, expounding upon their meaningfulness in an exhaustive fashion.

Although the article is quite well-written, I would have it proofread by a native speaker of English, in order to improve flow and readability overall, in addition to some minor grammar/vocabulary flaws.

This is a remarkably well-done research article and praiseworthy effort overall, hence I will be looking forward to a newly improved version.

Response:  Thanks for the constructive comments. The perspective of the findings are list in the discussion section.

The shortage of the COVID-19 vaccines and ultralow temperature storage limited the global vaccine distribution. The subunit vaccine is safe and stored in the freezer. Although the vaccine manufacturing capacity is limited in the present, the reduction of the antigen doses can increase the supply of the vaccine. Current subunit COVID-19 vaccines contained 5-15 μg antigen [9-10]. If the antigen amount can be reduced to 1/10, the vaccine can increase 10-fold supply. These indicated that adjuvant is critical in the vaccine supply and resolve the shortage of the vaccines. The antigen design and selection of adjuvants are important for the emerging infectious diseases or routing influenza vaccines vaccination.

References:

  1. Roberts, A.; Vogel, L.; Guarner, J.; Hayes, N.; Murphy, B.; Zaki, S.; Subbarao, K., Severe acute respiratory syndrome coronavirus infection of golden Syrian hamsters. Journal of virology 2005, 79 (1), 503-11.
  2. Liu, Y.; Hu, G.; Wang, Y.; Ren, W.; Zhao, X.; Ji, F.; Zhu, Y.; Feng, F.; Gong, M.; Ju, X.; Zhu, Y.; Cai, X.; Lan, J.; Guo, J.; Xie, M.; Dong, L.; Zhu, Z.; Na, J.; Wu, J.; Lan, X.; Xie, Y.; Wang, X.; Yuan, Z.; Zhang, R.; Ding, Q., Functional and genetic analysis of viral receptor ACE2 orthologs reveals a broad potential host range of SARS-CoV-2. Proceedings of the National Academy of Sciences of the United States of America 2021, 118 (12).
  3. Sia, S. F.; Yan, L. M.; Chin, A. W. H.; Fung, K.; Choy, K. T.; Wong, A. Y. L.; Kaewpreedee, P.; Perera, R.; Poon, L. L. M.; Nicholls, J. M.; Peiris, M.; Yen, H. L., Pathogenesis and transmission of SARS-CoV-2 in golden hamsters. Nature 2020, 583 (7818), 834-838.
  4. Chan, J. F.; Zhang, A. J.; Yuan, S.; Poon, V. K.; Chan, C. C.; Lee, A. C.; Chan, W. M.; Fan, Z.; Tsoi, H. W.; Wen, L.; Liang, R.; Cao, J.; Chen, Y.; Tang, K.; Luo, C.; Cai, J. P.; Kok, K. H.; Chu, H.; Chan, K. H.; Sridhar, S.; Chen, Z.; Chen, H.; To, K. K.; Yuen, K. Y., Simulation of the Clinical and Pathological Manifestations of Coronavirus Disease 2019 (COVID-19) in a Golden Syrian Hamster Model: Implications for Disease Pathogenesis and Transmissibility. Clinical infectious diseases : an official publication of the Infectious Diseases Society of America 2020, 71 (9), 2428-2446.
  5. Klein, S. L.; Flanagan, K. L., Sex differences in immune responses. Nature reviews. Immunology 2016, 16 (10), 626-38.
  6. Klein, S. L.; Marriott, I.; Fish, E. N., Sex-based differences in immune function and responses to vaccination. Trans R Soc Trop Med Hyg 2015, 109 (1), 9-15.
  7. Pellini, R.; Venuti, A.; Pimpinelli, F.; Abril, E.; Blandino, G.; Campo, F.; Conti, L.; De Virgilio, A.; De Marco, F.; Di Domenico, E. G.; Di Bella, O.; Di Martino, S.; Ensoli, F.; Giannarelli, D.; Mandoj, C.; Manciocco, V.; Marchesi, P.; Mazzola, F.; Moretto, S.; Petruzzi, G.; Petrone, F.; Pichi, B.; Pontone, M.; Zocchi, J.; Vidiri, A.; Vujovic, B.; Piaggio, G.; Morrone, A.; Ciliberto, G., Initial observations on age, gender, BMI and hypertension in antibody responses to SARS-CoV-2 BNT162b2 vaccine. EClinicalMedicine 2021, 36, 100928.
  8. Terpos, E.; Trougakos, I. P.; Apostolakou, F.; Charitaki, I.; Sklirou, A. D.; Mavrianou, N.; Papanagnou, E. D.; Liacos, C. I.; Gumeni, S.; Rentziou, G.; Korompoki, E.; Papassotiriou, I.; Dimopoulos, M. A., Age-dependent and gender-dependent antibody responses against SARS-CoV-2 in health workers and octogenarians after vaccination with the BNT162b2 mRNA vaccine. Am J Hematol 2021, 96 (7), E257-E259.
  9. Sacks, H. S., The Novavax vaccine had 90% efficacy against COVID-19 >/=7 d after the second dose. Ann Intern Med 2021, 174 (11), JC124.
  10. Hsieh, S. M.; Liu, M. C.; Chen, Y. H.; Lee, W. S.; Hwang, S. J.; Cheng, S. H.; Ko, W. C.; Hwang, K. P.; Wang, N. C.; Lee, Y. L.; Lin, Y. L.; Shih, S. R.; Huang, C. G.; Liao, C. C.; Liang, J. J.; Chang, C. S.; Chen, C.; Lien, C. E.; Tai, I. C.; Lin, T. Y., Safety and immunogenicity of CpG 1018 and aluminium hydroxide-adjuvanted SARS-CoV-2 S-2P protein vaccine MVC-COV1901: interim results of a large-scale, double-blind, randomised, placebo-controlled phase 2 trial in Taiwan. Lancet Respir Med 2021, 9 (12), 1396-1406.

Round 2

Reviewer 2 Report

Dear Authors,

The manuscript has been improved and the points raised in my review have been addressed to a satisfactory degree.

Sincerely,